# Evaluation of antiviral T cell responses and T$_{SCM}$ cells in volunteers enrolled in a phase I HIV-1 subtype C prophylactic vaccine trial in India

Sivasankaran Munusamy Ponnan[1], Peter Hayes[2], Natalia Fernandez[2], Kannan Thiruvengadam[1], Sathyamurthi Pattabiram[1], Manohar Nesakumar[1], Ashokkumar Srinivasan[1], Sujitha Kathirvel[1], Janani Shankar[1], Rajat Goyal[3], Nikhil Singla[3], Joyeeta Mukherjee[3], Shweta Chatrath[3], Jill Gilmour[2], Sudha Subramanyam[1], Srikanth Prasad Tripathy[1], Soumya Swaminathan[1], Luke Elizabeth Hanna◉[1]¤*

1 National Institute for Research in Tuberculosis (Indian Council of Medical Research), Chennai, India,
2 IAVI Human Immunology Laboratory, Imperial College, London, England, United Kingdom, 3 International AIDS Vaccine Initiative, New Delhi, India

¤ Current address: Department of HIV/AIDS, National Institute for Research in Tuberculosis, Chennai, India
* hanna@nirt.res.in

**Data Availability Statement:** All relevant data are within the paper and its Supporting Information files.

## Abstract

T cells play an important role in controlling viral replication during HIV infection. An effective vaccine should, therefore, lead to the induction of a strong and early viral-specific CD8$^+$ T cell response. While polyfunctional T cell responses are thought to be important contributors to the antiviral response, there is evidence to show that polyfunctional HIV- specific CD8$^+$ T cells are just a small fraction of the total HIV-specific CD8$^+$ T cells and may be absent in many individuals who control HIV replication, suggesting that other HIV-1 specific CD8$^+$ effector T cell subsets may be key players in HIV control. Stem cell-like memory T cells (T$_{SCM}$) are a subset of T cells with a long half-life and self-renewal capacity. They serve as key reservoirs for HIV and contribute a significant barrier to HIV eradication. The present study evaluated vaccine-induced antiviral responses and T$_{SCM}$ cells in volunteers vaccinated with a subtype C prophylactic HIV-1 vaccine candidate administered in a prime-boost regimen. We found that ADVAX DNA prime followed by MVA boost induced significantly more peripheral CD8$^+$ T$_{SCM}$ cells and higher levels of CD8$^+$ T cell-mediated inhibition of replication of different HIV-1 clades as compared to MVA alone and placebo. These findings are novel and provide encouraging evidence to demonstrate the induction of T$_{SCM}$ and cytotoxic immune responses by a subtype C HIV-1 prophylactic vaccine administered using a prime-boost strategy.

## Importance of the study

Effector and memory T cells play a crucial role in HIV infection and are well-preserved in controllers, but become exhausted during infection. Stem cell like memory (T$_{SCM}$) cells are a subset of T cells that possess long half-life, self-renewal capacity and contribute a significant

**Funding:** The vaccine trial and the present study were supported and coordinated by the International AIDS Vaccine Initiative (IAVI). IAVI's work is made possible by generous support from many donors including the Bill & Melinda Gates Foundation, the Ministry of Foreign Affairs of Denmark, Irish Aid, the Ministry of Finance of Japan in partnership with The World Bank, the Ministry of Foreign Affairs of the Netherlands, the Norwegian Agency for Development Cooperation (NORAD), the United Kingdom Department for International Development (DFID), and the USAID. The full list of IAVI donors is available at www.iavi. org. This study was possible through the generous support of the American people through funding from the United States Agency for International Development (USAID; Grant ID: 2233). The contents are the responsibility of the authors and do not necessarily reflect the views of USAID or the United States Government.

**Competing interests:** NO authors have competing interests

barrier to HIV eradication. The present study demonstrated increased CD8$^+$ T cell mediated viral inhibition of different clades of HIV-1 viruses and higher frequencies of peripheral T$_{SCM}$ cells in volunteers who received a subtype C prophylactic HIV-1 vaccine candidate that was tested in a Phase I trial in India. Our findings indicate that the vaccine candidate and vaccination regimen used in the trial were capable of eliciting a favourable antiviral response and inducing production of CD8$^+$ T cells and T$_{SCM}$ cells that offer further promise for future HIV vaccine research.

## Introduction

HIV/AIDS continues to be a significant challenge for global public health even after 35 years of its discovery, with HIV's sequence diversity and rapid evolution being major obstacles for successful vaccine development [1]. Though several significant advances have been made with respect to the design and development of HIV vaccines, there have not been any major breakthroughs in the field [2]. To date, only one candidate vaccine tested in the RV144 trial in Thailand, demonstrated the ability to significantly reduce the risk of HIV-1 acquisition, giving a modest protective efficacy of 31%, providing new hope for antibody-based vaccine development [3].

There is strong evidence to suggest that T cell-mediated responses play an important role in suppression of virus replication and clearance of infected cells from the host [4–6]. Several cohort studies have reported that generation and proliferation of HIV-specific CD8$^+$ T cells correlate inversely with viral load [7–10]. Thus, CD8$^+$ T cell-mediated immune responses are clearly capable of modulating the course of HIV infection and therefore constitute an important dimension for vaccine research. Persistence and maintenance of the CD8$^+$ T cell response are dependent on CD4$^+$ T cell help [11]. However, a recent study reported that naturally activated CD8$^+$ T cells may persist in the absence of CD4$^+$ T cells [12] and that a persistent memory CD8$^+$ T cell response is important for viral control [13–17]. Long-lived memory CTLs can directly kill cells infected with intracellular pathogens by producing cytokines, chemokines (IL-2, IFN-γ, TNF-α, MIP-1) and cytotoxins (perforin, granzymes and granulysin). CTLs can also use the Fas ligand that binds to the Fas receptors present on the surface of many cell types and induce cellular apoptosis [18].

Recent studies have identified a subset of memory T cells that possess stem cell-like properties, called stem-like memory T cells (T$_{SCM}$) [19–21]. T$_{SCM}$ cells are the least differentiated among the distinct memory populations. They differentiate into different types of memory T cells that play vital roles in controlling the immune response [19]. T$_{SCM}$ cells have been well characterized in cancers and other autoimmune diseases. CD8$^+$ T$_{SCM}$ cells have been proposed as therapeutic targets in acquired aplastic anaemia [22]. However, the role of CD4$^+$ and CD8$^+$ T$_{SCM}$ cells in HIV infection and vaccination, and their contribution to antiviral immune defence remains unclear. Earlier studies reported that the frequency of CD8$^+$ memory T cells (central and effector memory cells) correlates with lower viral load in chronic HIV infection [23–25] and that lower levels of expression of PD-1 on these cells [26–28] indicate that long-lived memory cells may be derived from T$_{SCM}$ cells. Thus, measuring vaccine-induced T$_{SCM}$ cells could provide key insights into the ability of the vaccine to elicit robust and persistent cellular immune responses.

Characterization of the nature of the antiviral response elicited by vaccination would provide the most critical clues to understanding the exact mechanisms that contribute to vaccine efficacy. Assays that measure CD8$^+$ T cell-mediated inhibition of replication of diverse HIV-1 isolates (viral inhibition assay/VIA) may be used to assess the breadth of the direct antiviral function of these cells [29].

Previously, we examined vaccine-induced humoral and cell-mediated responses to the same subtype C HIV-1 vaccine (ADVAX DNA prime and MVA boost) tested in a phase I clinical trial in India (P001 trial) and reported encouraging findings [30, 31]. In the current study, we investigated vaccine-induced peripheral $T_{SCM}$ cells and *in vitro* viral inhibitory capacity of vaccine-induced CD8$^+$ T cells in the same cohort.

## Materials and methods

### Samples

The study was performed using cryopreserved peripheral blood mononuclear cells (PBMC) obtained from volunteers who participated in the IAVI sponsored P001 trial conducted during the period 2009–2010 [31]. The trial had enrolled a total of 32 HIV-uninfected healthy volunteers at two sites in India: 16 volunteers at the National Institute for Research in Tuberculosis (NIRT), Chennai, and 16 volunteers at the National AIDS Research Institute (NARI), Pune. The present analysis was confined to samples stored at NIRT alone and focuses on Chennai participants. The 16 individuals (9 males and 7 females) were randomly assigned to either Group A or B, with eight participants in each Group. Group A participants received two intramuscular (I.M.) injections of ADVAX or placebo at baseline (time '0') and 1 month, followed by two I.M. injections of TBC-M4 or placebo at months 3 and 6. Group B participants received three I.M. injections of TBC-M4 or placebo at time 0, months 1 and 6. Among the 8 volunteers in each Group, 6 received the vaccine and 2 received placebo (Fig 1).

### Candidate vaccines

ADVAX is a DNA vaccine formulated by the Aaron Diamond AIDS Research Center (ADARC), New York, USA, and manufactured by Vical Inc., San Diego, CA, USA (Lot# 04030248), utilizing the structural features of the commercial plasmid backbone pVAX1 [32, 33]. The vaccine contains two plasmid constructs (one plasmid cloned with the coding sequences of *gag* and *env* genes of the Chinese HIV-1 clade B/C strain and the other one cloned with the coding sequences of *nef/tat* and *pol*, intended to express a fusion protein)

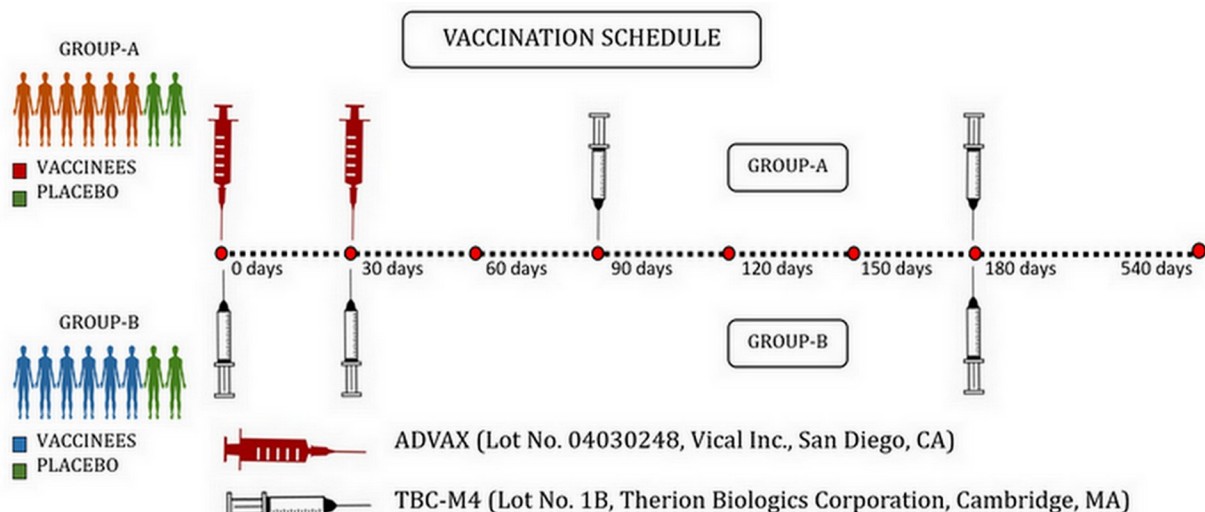

**Fig 1. Vaccination schedule followed in the IAVI Phase-I prime-boost HIV-1 subtype C prophylactic vaccine trial (P001 trial).**

mixed in a 1:1 ratio. The candidate vaccine was suspended in sterile phosphate-buffered saline with 0.01M sodium phosphate and 150 mM sodium chloride and was formulated to contain 4 mg of the DNA in a total volume of 1mL.

TBC-M4 is a vaccine manufactured by Therion Biologics Corporation, Cambridge, MA, USA (Lot# 1B). It is a recombinant Modified Vaccinia Ankara (MVA) virus containing the coding sequences of *gag*, *env*, *reverse transcriptase (RT)*, *tat*, *rev* and *nef* of an Indian HIV-1 clade C strain [34]. The vaccine was suspended in phosphate-buffered saline and 10% glycerol to contain $5\times10^6$ PFU in a total volume of 0.5 mL. The amino acid sequence homology between the two vector inserts was found to be greater than 85% (ENV: 87.1%; GAG: 95%; POL/RT: 96.4%).

### Virus inhibition assay

**Viruses.** HIV-1 subtype A (U455), subtype B (IIIB), subtype C (247FV2), subtype D (CBL4) and subtype AD (ELI) isolates were used in the VIA. U455 and CBL-4 were obtained from NIBSC, UK, ELI and IIIB were obtained from the NIH AIDS Research and Reference Reagent Program (Bethesda, MD), 247FV2 was kindly provided by George Shaw (University of Pennsylvania, PA). The titer of each viral isolate was determined by end-point dilution and defined as 50% tissue culture infectious dose ($TCID_{50}$) using the TZM-bl cell line [29] obtained from the NIH AIDS Research and Reference Reagent Program.

**Expansion of $CD4^+$ and $CD8^+$ T cells.** PBMC were thawed and resuspended at $1–1.5\times10^6$ cells/ml in RPMI containing 10% Fetal Bovine Serum (FBS), 50 units IL-2 (R10/50) and 0.5μg/ml $CD3^+$/$CD4^+$ or $CD3^+$/$CD8^+$ bi-specific antibody for expansion of $CD8^+$ and $CD4^+$ T cell sub-populations respectively. Culture volumes were doubled at days 3 and 6 with R10/50 medium. Overall, the cell numbers increased by a factor of mean 5, and the typical purity of day 7 cultures was 97 and 87% for $CD4^+$ and $CD8^+$ T cells respectively, demonstrating positive expansion and enrichment ($\geq$90% of the CD3+ T cells in culture) of the required T cell sub-population [29].

**Viral inhibition assay.** Viral inhibition assay was performed as described by Spentzou et al., 2010. Briefly, separate cultures of 7 day-expanded $CD4^+$ T cells infected with different HIV-1 isolates at a multiplicity of infection (MOI) of 0.01 were established. To limit variation due to the possible effects of the vaccine regimen on $CD4^+$ target cells, a single population of target $CD4^+$ T cells was generated for each subject; wherever available, this cell population was generated from the baseline pre-vaccination sample. Virus-infected target cells were co-cultured with autologous $CD8^+$ T cells obtained at the pre-vaccination time point and at two post-vaccination time points. $CD4^+$ and $CD8^+$ T cells were co cultured in 1:1 ratio. In our assay, 0.5 million viral infected $CD4^+$ T cells were co cultured with 0.5 million $CD8^+$ T cells. Every 3 to 4 days, half of the culture supernatant was removed and replaced with fresh R10/50 medium and assessed for Gag p24 content using a commercially available enzyme-linked immunosorbent assay (ELISA) kit (PerkinElmer, United Kingdom). $CD8^+$ T cell-mediated inhibition was expressed as $log_{10}$ reduction in the p24 content of day 13 supernatants from $CD8^+$ and $CD4^+$ T cell co-cultures as compared to that of CD4+ T cells alone.

Cut-offs were defined by the 97.5th percentile of the baseline VIA response as estimated using PROC QUANTREG in SAS 9.2 (SAS Institute Inc., Cary, NC). VIA response for each virus was defined as positive if the following three criteria were fulfilled: (i) $log_{10}$ inhibition is greater than 1.5 for all HIV-1 isolates, (ii) pre-vaccination response for the same virus is negative, and (iii) the difference between the post-vaccination and pre-vaccination response is $\geq$0.6 $log_{10}$ inhibition [29].

## Polychromatic flow cytometric analysis of memory T cells and $T_{SCM}$ cells

Cryopreserved PBMC obtained at pre-vaccination, on the day of second MVA vaccination (VAC II), one week after second vaccination, on the day of last MVA vaccination (VAC III) and 1, 2 and 48 weeks after last MVA vaccination, were analyzed for memory T cell subset phenotypes. PBMC at pre-vaccination, one week after VAC II, one week after VAC III, and 2 and 48 weeks after last MVA vaccination were analyzed for $T_{SCM}$ cell phenotype. Two million PBMC were washed with FACS buffer and stained with Live/Dead Fixable Aqua blue dye (Invitrogen). The following cocktail of monoclonal antibodies were used to enumerate the different cell types: T Memory and $T_{SCM}$ cell panel: anti-CD3-APCH7, anti-CD122-PE, anti-CD4-PERCP, anti-CD28-FITC, anti-CD95-PECF594, anti-CD45RA-APC, anti-CCR7-PECY7, and anti-CD8-APCR700. All antibodies were procured from Becton-Dickinson, India (S1 Table). Cells were stained for 30 minutes in the dark, washed and fixed with 4% paraformaldehyde for 5 minutes, and acquired on a FACS ARIA SORP flow cytometer (Becton Dickinson). A minimum of 1,000,000 total events were acquired and data were analyzed using FlowJo software, version 10.4 (Treestar, Ashland, OR).

The following panels were used to identify the different memory subsets [19]. Naïve cells—CD45RO- CCR7+ CD28+ CD95-, Stem cell-like memory cells—CD45RO- CCR7+ CD28 + CD95+, Central memory cells—CD45RO+ CCR7+ CD28+ CD95+, Effector memory cells—CD45RO+ CCR7- CD28- CD95+, Terminal Memory cells—CD45RO+CCR7- CD28+ CD95 +, and terminal effector cells—CD45RO-CCR7- CD28- CD122-CD95+.

## Statistics

Statistical analyses were performed using GraphPad Prism, version 7.05 (GraphPad Software, Inc., CA). Values are presented as median and interquartile ranges. The percentage frequency of Memory T cells and $T_{SCM}$ cells were compared within the placebo and Group A and Group B vaccinees at different time points using Kruskal-Wallis test, followed by subgroup analysis using Dunn's multiple comparison test to identify differences between the groups. For all analyses, differences were considered significant if p value was <0.05.

## Ethics statement

The present study was approved by the Institutional Ethics Committee of NIRT (NIRT IEC No. 2015013). During the trial, written statements of informed consent were obtained from the study volunteers. The trial was supervised by the research personnel of the International AIDS Vaccine Initiative (IAVI) and the study was conducted in accordance with the ethical principles stated in the Declaration of Helsinki [16], Guidelines for Good Clinical Practice (GCP) framed in the International Conference on Harmonization (ICH), and Good Clinical Laboratory Practice (GCLP) outlined by the Research Quality Association (RQA), UK, were adhered to during the conduct of the study.

# Results

## Assessment of viral inhibition activity

Viral inhibition activity was assessed with five HIV-1 isolates belonging to different clades, at three-time points (baseline, two weeks post 1st MVA vaccination and two weeks post last MVA vaccination). VIA response was absent in the placebo group at all-time points and in the vaccinated individuals at baseline/pre-vaccination. Positive VIA responses were observed in Group A and B vaccinees post-vaccination. The VIA responses were more frequent in Group A as compared to Group B at two weeks post 1st MVA vaccination; 4 of the 6 Group A

volunteers inhibited the U455 isolate (>1.5log inhibition), 2 volunteers inhibited 247FV2, one individual inhibited IIIB and one individual inhibited ELI-1 with $\log_{10}$ reduction in p24 level of >1.5. In contrast, only one individual in Group B demonstrated VIA activity against the U455 isolate. At two weeks post last MVA vaccination, Group A volunteers exhibited VIA activity against all 5 HIV isolates tested; 3 individuals inhibited the U455 isolate, 2 individuals inhibited IIIB, and 3 individuals inhibited the 247FV2, CBL4 and ELI-1 isolates at >1.5log. However, in Group B, VIA activity was seen only against two of the HIV isolates; 2 volunteers inhibited U455 and 1 volunteer inhibited CBL4 at >1.5log. Overall, we observed much better VIA responses in group A volunteers as compared to group B and placebo. These observations suggest that the heterologous prime-boost vaccine regimen was more potent in eliciting a CD8+ T cell-dependent antiviral response against different HIV isolates (Fig 2).

## Induction of memory T cells subsets by vaccination

PBMC were analysed by multicolour flow cytometry to determine vaccine-induced alterations in the frequency of memory T cell subsets [central memory T cells (CM), effector memory T cells (EM), terminal effector T cells (TE), and naïve T cells (TN)]. Among CD4+ T cells, CM cells were defined as CD3+CD4+CD45RO+CCR7+, EM cells as CD3+CD4+CD45RO+CCR7-, TE cells as CD3+CD4+CD45RO-CCR7- and TN cells as CD3+CD4+CD45RO-CCR7+. Among the CD8+ T cells, CM cells were defined as CD3+CD8+CD45RO+CCR7+, EM cells as CD3+CD8+CD45RO+CCR7-, TE cells as CD3+CD8+CD45RO-CCR7- and TN cells as CD3+CD8+CD45RO-CCR7+ (S1 Fig).

The proportion of CD4+ TN cells was found to be lower in Groups A and B at 1st week post VAC-II, VAC-III, 1st week post VAC-III and 2nd week post VAC-III time points as compared to the placebo. Similarly, decreased frequency of CD8+ TN cells was observed in the vaccine

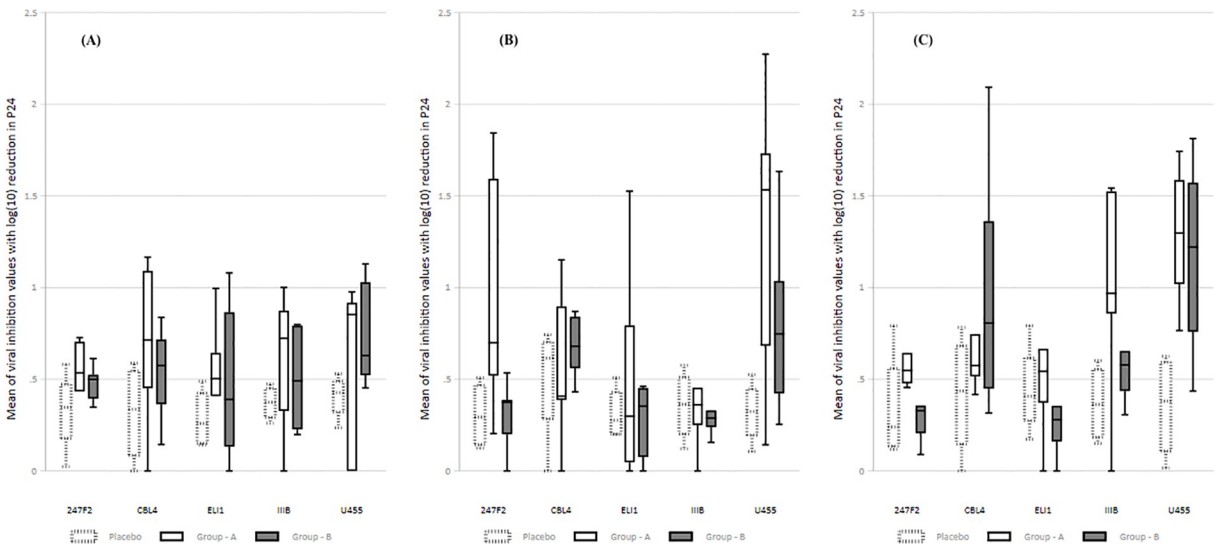

**Fig 2. Viral inhibition activity in terms of $\log_{10}$ reduction in p24 level at pre-vaccination, two weeks post first MVA vaccination and two weeks post last MVA vaccination against the five HIV-1 subtypes: 247FV2 (clade C), CBL-4 (clade D), ELI (clade A/D), IIIB (clade B) and U455 (clade A).** (A) Viral inhibition activity at the pre-vaccination time point. (B) Viral inhibition activity at two weeks post first MVA vaccination. (C) Viral inhibition activity at two weeks post last MVA booster. The box and whisker plots summarize the distribution of positive responses (median, 1st and 3rd quartile). The y-axis represents $\log_{10}$ reduction in P24 antigen level. Shaded boxes represent placebo values, white boxes represent Group A and black boxes represent group B values. The mean $\log_{10}$ values were compared at different time points within the 3 groups for the 5 different viral clades using Kruskal-Wallis test, followed by subgroup analysis using Dunn's multiple comparison test to identify differences between the three groups. For all analyses, differences were considered significant if p value was <0.05.

recipients at post-vaccination time points (S2 and S3 Tables). In contrast, the frequency of CD4+ CM cells was higher in the vaccinated groups at VAC-II, 1st week post VAC-II and 1st week post VAC-III time points as compared to the control group, while there was no difference in the frequency of CD8+ CM cells in any of the three groups.

We observed that both group A and group B vaccines had significantly higher TE cell frequencies in the CD4+ as well as CD8+ T cell compartments at all post-vaccination time points as compared to the placebo subjects. The increase in CD8+ TE cells was found to be significant in Group A at all post-vaccination time points as compared to group B (p = 0.003 to 0.022) (Fig 3; S2 Table). In the CD4 compartment, the frequency of TE cells was found to be significantly higher in group A than in group B at VAC-III and 1st week post VAC-III time points only (p = 0.009; p = 0.001) (Fig 4; S3 Table).

Similarly, significantly elevated numbers of EM T cells was seen in Group A as compared to Group B and placebo controls. The increase in CD8+ EM cells was found to be significantly higher in Group A at all post-vaccination time points when compared to placebo as well as group B (Group A vs Placebo: p<0.001 to 0.009; Group A vs group B: p = 0.002 to 0.039) (Fig 3; S2 Table). CD4+ EM cells were significantly higher in group A at all post-vaccination time points except VAC III and 2nd week post VAC-III as compared to placebo (p = 0.019 to 0.007). However, when compared to group B, the increase was significant only at the VAC-II time point (p = 0.019). Group B volunteers had increased frequency of CD4+ EM cells at 1st week post VAC-II, 2nd week post VAC-III and 48th week post VAC-III as compared to placebo (p = 0.002; p = 0.0006; p = 0.008 respectively) (Fig 4; S3 Table). Difference in the number of total CD8+ and CD4+ memory T cell subsets is presented in S4 and S5 Tables respectively. These observations indicate that both the heterologous and homologous prime-boost vaccination regimens resulted in increased frequencies of peripheral effector and memory CD4+ and CD8+ T cells, with the increase being significantly higher in group A as compared to group B.

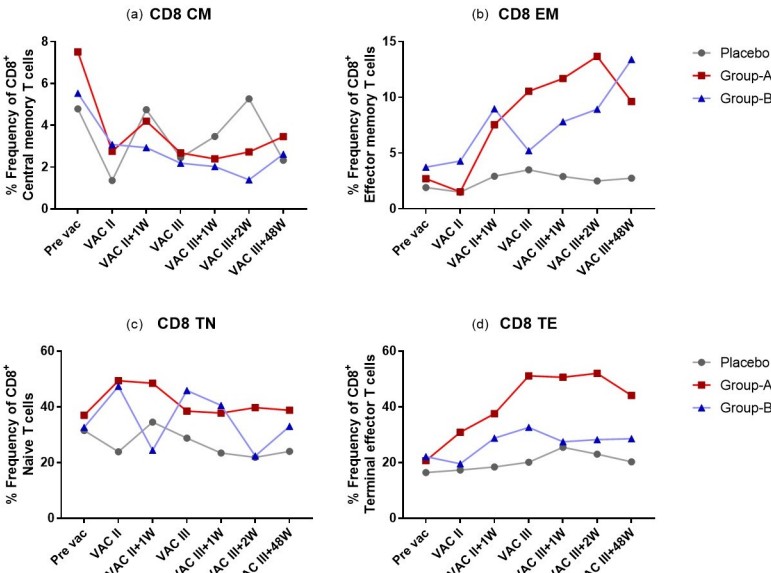

**Fig 3. CD8+ memory T cell subsets in placebo, Group A and Group B volunteers.** Frequency (%) of total CD8+ memory T cell subsets in Placebo, Group A and Group B volunteers at Pre-vaccination, one week post VAC-II, and one, two and 48 weeks post VAC-III. The superimposed dot plots summarize the % frequency of total vaccine-induced CD8+ memory T cell subsets.

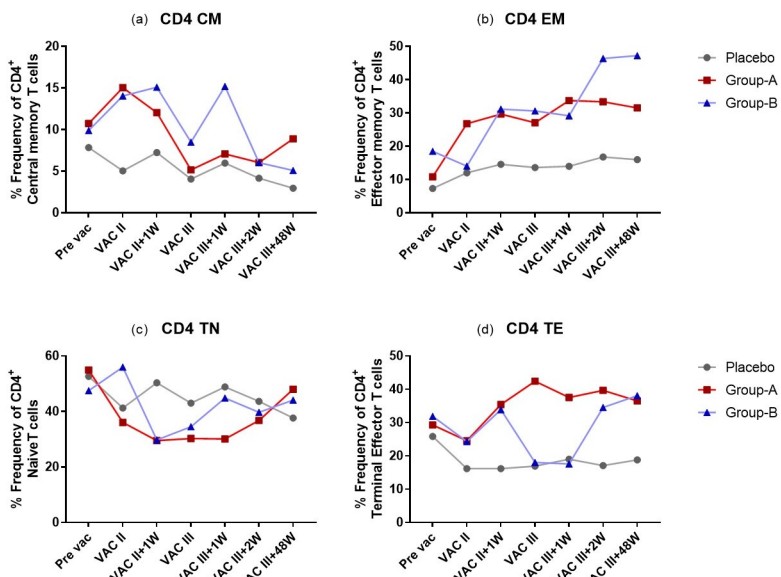

**Fig 4. CD4+ Memory T cell subsets in placebo, Group A and Group B volunteers.** Frequency (%) of total CD4+ memory T cell subsets in Placebo, Group A and Group B at Pre-vaccination, one week post VAC-II, and one, two and 48 weeks post VAC-III. The superimposed dot plots summarize the % frequency of total CD4+ memory T cell subsets.

## Induction of $T_{SCM}$ cells by vaccination

CD4+ $T_{SCM}$ cells were defined as CD3+CD4+CD45RO-CCR7+CD28+CD95+ cells and CD8 + $T_{SCM}$ cells were defined and identified as CD3+CD8+CD45RO-CCR7+CD28+CD95+ cells (S1 Fig). The percentage of circulating CD4+ or CD8+ $T_{SCM}$ cells were relatively small (0.15 to 0.22%) in the placebo group as well as at baseline in the vaccinated groups. Vaccination resulted in a significant increase in the frequency of $T_{SCM}$ cells in both CD4+ and CD8+ T cell compartments in both Groups A and B (Table 1A; Figs 5 and 6). CD8+ $T_{SCM}$ cell frequency was significantly higher in Group A at all post-vaccination time points as compared to the placebo group (p<0.001 to 0.003), at one week post VAC-II and III, and 48 weeks post VAC-III, as compared to Group B (p = 0.005 to 0.025) (Fig 6). CD4+ $T_{SCM}$ cell frequencies were significantly higher in Group A at one week post VAC-II and one week post VAC-III time points as compared to placebo (p≤0.002), and at one week post VAC-II when compared to Group B volunteers (p = 0.025) (Table 1B; Fig 6). There was a clear trend towards increased frequencies of this sub-population of cells in Group A volunteers during the period of vaccination (Fig 5), suggesting that the ADVAX/MVA prime-boost regimen induced high frequencies of CD8+ and CD4+ $T_{SCM}$ cells.

## Discussion

The ultimate solution to the HIV epidemic is a vaccine that will substantially reduce transmission [35–37]. The risk of transmission is associated with high levels of viremia seen in acute and uncontrolled chronic infection. Though several studies have documented an inverse correlation between activated immune cells and viral load [3, 38, 39], their successful incorporation into vaccine design has not been achieved so far [37, 40]. Adaptive T cell immunity provides the individual with a specialized defence against intracellular pathogens. CD8+ T cells were first implicated in suppressing HIV replication when reductions in viral load were found to correlate with the appearance of HIV-specific CD8+ T cells [41, 42]. Subsequent studies

**Table 1. Induction of $T_{SCM}$ cells by vaccination.** A) Comparison of mean frequency (%) of CD4$^+$ $T_{SCM}$ cells in Placebo, Group A and Group B volunteers. B) Comparison of mean frequency of CD8$^+$ $T_{SCM}$ cells in Placebo, Group A and Group B volunteers.

| A | | | | | |
|---|---|---|---|---|---|
| **Time** | **CD4$^+$ $T_{SCM}$ cells** | | | **Sig.*** | **Sub-group analysis** |
| | **Placebo P (n = 4)** | **Group A (n = 6)** | **Group B (n = 6)** | | |
| | **median (IQR)** | **median (IQR)** | **median (IQR)** | | |
| Pre-VAC | 0.15 (0.11–0.18) | 0.19 (0.15–0.22) | 0.18 (0.15–0.21) | 0.247 | - |
| VAC-II+1W | 0.18 (0.16–0.21) | 1.38 (1.02–1.65) | 0.77 (0.52–0.97) | 0.005 | A vs. P (<0.001); B vs. P 0.025) |
| VAC-III+1W | 0.16 (0.09–0.20) | 1.37 (0.46–1.63) | 0.69 (0.34–0.91) | 0.018 | A vs. P (0.002) |
| VAC-III+2W | 0.19 (0.16–0.21) | 1.16 (0.19–1.32) | 0.19 (0.17–0.26) | 0.191 | - |
| VAC-III+48W | 0.14 (0.13–0.24) | 0.32 (0.15–0.76) | 0.26 (0.18–0.42) | 0.118 | - |
| **B** | | | | | |
| **Time** | **CD8$^+$ $T_{SCM}$ cells** | | | **Sig.*** | **Sub-group analysis** |
| | **Placebo P (n = 4)** | **Group A (n = 6)** | **Group B (n = 6)** | | |
| | **median (IQR)** | **median (IQR)** | **median (IQR)** | | |
| Pre-VAC | 0.17 (0.13–0.27) | 0.22(0.17–0.32) | 0.18 (0.13–0.24) | 0.428 | - |
| VAC-II+1W | 0.21 (0.08–0.26) | 1.38 (0.87–1.89) | 0.96 (0.60–1.05) | 0.008 | A vs. P (0.001); B vs. P (0.017) |
| VAC-III+1W | 0.15 (0.13–0.17) | 1.77 (1.15–2.06) | 0.57 (0.43–1.76) | 0.005 | A vs. P (<0.001); B vs. P (0.025) |
| VAC-III+2W | 0.19 (0.19–0.26) | 2.92 (2.53–3.19) | 1.35 (0.59–1.99) | 0.003 | A vs. P (<0.001); A vs. B (0.014) |
| VAC-III+48W | 0.14 (0.07–0.16) | 1.58 (0.72–1.82) | 1.52 (0.77–1.79) | 0.014 | A vs. P (0.003); B vs. P (0.005) |

*K-Wallis test was performed to analyze differences within the individual groups, and Dunn's multiple comparison test was performed to identify differences between the 3 groups.

showed that long term non-progressors (LTNP) possess relatively high CTL and CD4$^+$ T cell proliferative responses despite low viral loads [42]. In SIV infection, removal of CTLs correlated with an increased life span of productively infected CD4$^+$ T cells and increased viral load, demonstrating clearly the importance of CTLs in viral control [43]. Ongoing efforts to develop an effective HIV vaccine are partly based on the principle that a specific antiviral CTL response is crucial for immune control of viral replication. This principle applies to various chronic and persistent viral infections such as hepatitis B, hepatitis C, CMV and Epstein-Barr virus [3].

The IAVI-sponsored Phase I P001 trial conducted in India explored the ability of an HIV-1 subtype C prophylactic vaccine candidate to influence the frequencies of different types of peripheral memory T cells (CM, EM and TE) and $T_{SCM}$ cells in volunteers. We observed that both the heterologous and homologous prime-boost vaccine regimens induced changes in proportions of CM, EM and TE cells in the CD4$^+$ as well as CD8$^+$ T cell compartments. Changes in the frequency of memory subsets were more pronounced in the CD8$^+$ T cell compartment as compared to the CD4$^+$ T cell compartment in both vaccinee groups, suggesting that both the ADVAX/MVA and MVA regimens were capable of eliciting memory CD8$^+$ T cells responses. These responses were maintained up to 48 weeks post last vaccination, implying that the vaccine-induced memory CD8$^+$ T cell response was long-lasting, and could potentially provide long term virological control.

$T_{SCM}$ cells are a subset of T cells that give rise to different populations of memory T cells (CM, EM and TE cells) [44]. Functional cellular responses of CMV, Flu, SIV and melanoma-specific CD8$^+$ $T_{SCM}$ have been previously reported [20]. Though the role of $T_{SCM}$ cells in HIV infection and antiviral response has not been completely elucidated, earlier studies have reported that elite controllers have a higher percentage of effector cells and immature memory HIV-1-specific CD8$^+$ T cells [45–47]. These studies suggest a role for HIV-1-specific $T_{SCM}$ cells in HIV-1 infection. Interestingly, the proportion of total CD8$^+$ $T_{SCM}$ cells was observed to

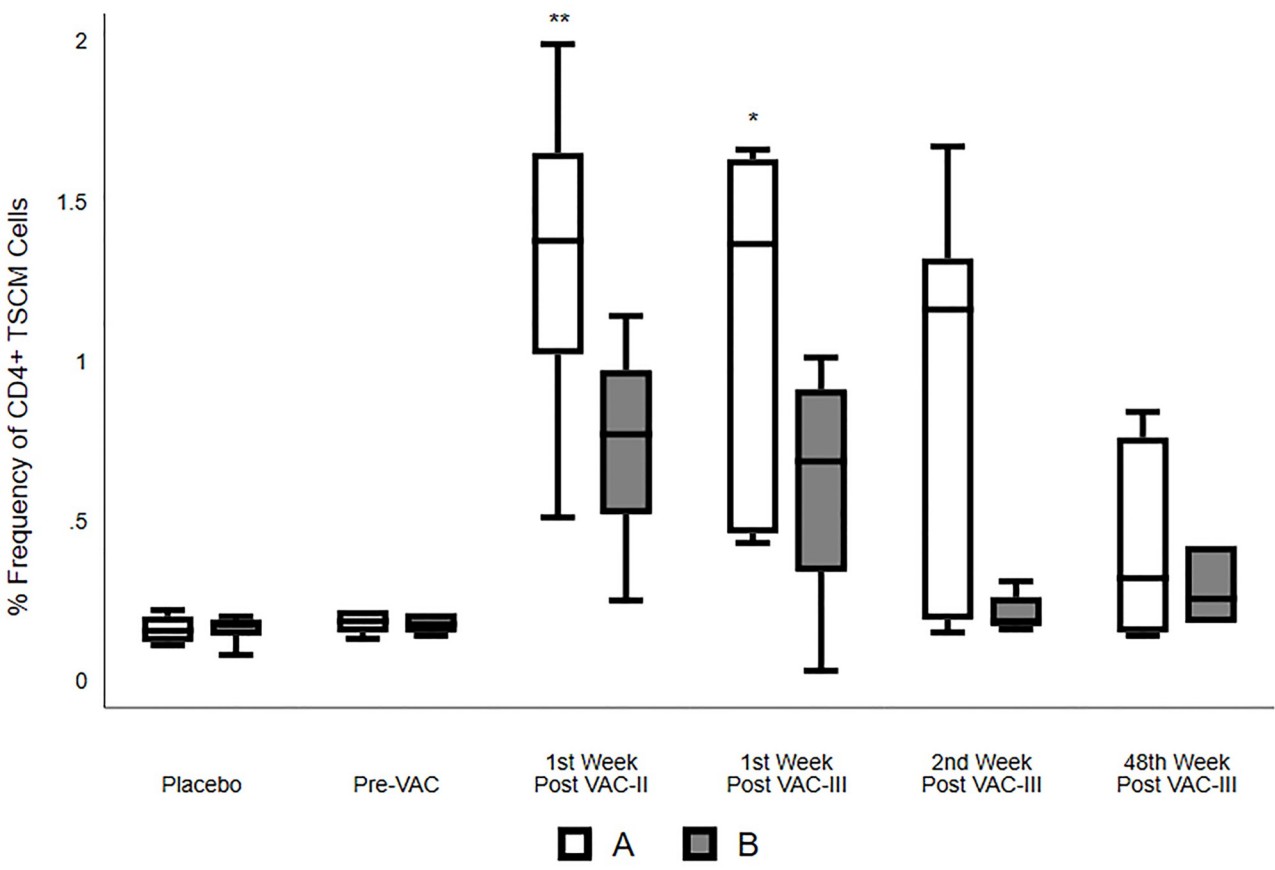

**Fig 5. CD4+ T_SCM cell subsets in placebo, Group A and Group B volunteers.** Frequency (%) of total circulating CD4+ T_SCM cells in Placebo, Group A and Group B at Pre-vaccination, one week post VAC-II, and one, two and 48 weeks post VAC-III. The box and whisker plots summarize the % frequency of circulating CD4+ T_SCM cells (median, 1st and 3rd quartiles). K-Wallis test was performed to analyze differences within the individual groups, and Dunn's multiple comparison test was performed to identify differences between the 3 groups. P—Placebo; A—Group A; B—Group B; *- p<0.05; **- p<0.01; **—p<0.001.

correlate inversely with levels of plasma viremia in untreated HIV-1 infected individuals [48]. On the other hand, CD4+ T_SCM cells, in addition to giving rise to different subsets of memory T cells, also constitute reservoirs for HIV-1 infection. Hence, it may be considered that the generation of CD8+ T_SCM cells may be a better correlate of protection against HIV infection, as these cells would not be infected with HIV-1. Our study identified a significantly higher proportion of circulating CD8+ T_SCM cells in volunteers who received the ADVAX and MVA or MVA alone as compared to placebo. The overall frequency of CD4+ and CD8+ T_SCM cells was higher in group A at two weeks post last MVA vaccination as compared to the pre-vaccination time point, suggesting that DNA priming and MVA boosting was more effective in inducing T_SCM cells and was capable of generating a long lasting memory and effector response.

CTLs are functionally defined as either polyfunctional CD8+ T cells that secrete multiple cytokines [49], or highly proliferative CD8+ T cells that kill HIV-infected target cells [50] and suppress HIV replication *in vitro* [51, 52]. Intracellular cytokine staining (ICS) and IFN-γ ELI-S_POT assays are routinely employed to evaluate vaccine-induced cellular immune responses. Though these responses do not correlate with *in vivo* virus control, they are helpful in demonstrating magnitudes and specificities of the T cell response. Hence, an assay that directly assesses the breadth of T cell-mediated antiviral activity against different HIV-1 isolates would

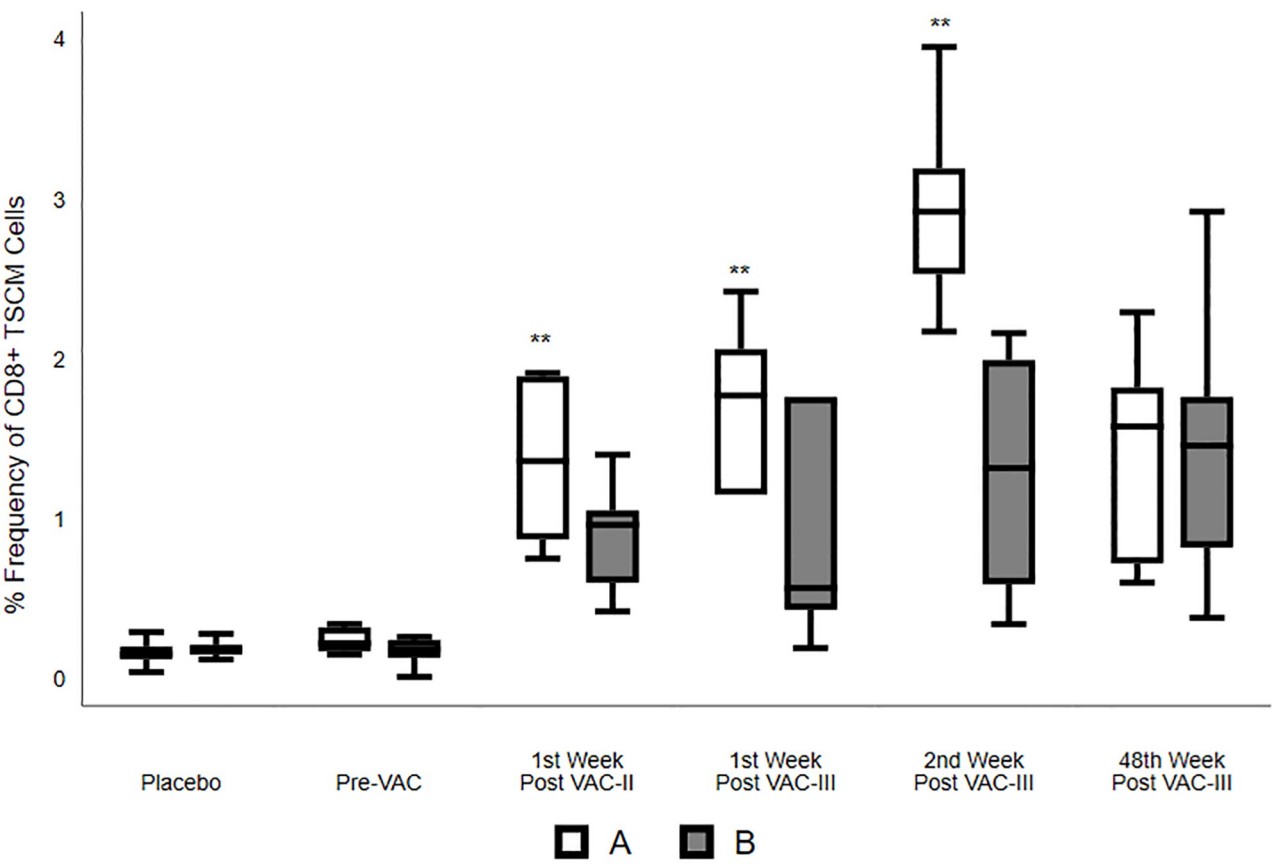

**Fig 6. CD8+ T_SCM cell subsets in placebo, Group A and Group B volunteers.** Frequency (%) of total circulating CD8+ T_SCM cells in Placebo, Group A and Group B at Pre-vaccination, one week post VAC-II, and one, two and 48 weeks post VAC-III. The box and whisker plots summarize the % frequency of circulating CD8+ T_SCM cells (median, 1st and 3rd quartiles). K-Wallis test was performed to analyze differences within the individual groups, and Dunn's multiple comparison test was performed to identify differences between the 3 groups. P—Placebo; A—Group A; B—Group B; *- $p < 0.05$; **- $p < 0.01$; ***- $p < 0.001$.

be best suited to evaluate the potential ability of the vaccine to induce a T cell response that can control virus *in vivo* [29].

Previously, we evaluated vaccine-induced T cell responses in these groups of individuals using IFN-γ ELIS_POT and ICS assays and reported increased IFN-γ ELIS_POT responses in Group A as compared to Group B [31, 53]. We also observed significantly elevated Env and Gag-specific mono- and bi-functional T cell responses in vaccinated individuals, with the magnitude of responses being significantly higher in Group A as compared to Group B [53]. In the present study, we analyzed the antiviral response of the vaccine-induced CD8+ T cells using the Virus Inhibition Assay. A similar vaccine trial, the P002 trial conducted in the UK with the same vaccine constructs, had reported that CD8+ T cell-mediated VIA activity was detected only in Group A against three HIV isolates tested: 247FV2 (clade C), 97ZA012 (clade C), and IIIB (clade B). Further, they reported an increase in the magnitude of the ICS response to HIV proteins in Group A as compared to Group B.

In the present study, CD8+ T cell-mediated VIA activity was studied against five HIV isolates, 247FV2 (clade C), ELI-1 (clade A/D), CBL4 (clade D), U455 (clade A) and IIIB (clade B). VIA activity was detected in Group A against four isolates: 247FV2, ELI-1, U455 and IIIB, whereas in Group B VIA activity was observed only against two isolates: U455 and IIIB. Four

volunteers exhibited viral inhibition activity at two weeks post first MVA vaccination and 5 volunteers showed viral inhibition activity at two weeks post last MVA vaccination in Group A, while in Group B only 3 volunteers showed VIA response at two weeks post last MVA vaccination alone. Our findings reveal elevated CD8[+] T cell-mediated VIA activity in terms of breadth as well as magnitude in Group A as compared to Group B. These observations support our earlier findings of significantly higher ICS and IFN-γ ELIS$_{POT}$ responses in Group A volunteers and also reflect the findings of the P002 vaccine trial. Findings of other studies also suggest that the quality of the immune response was enhanced by the DNA prime, when adenovirus or poxvirus vectors were used as the boost [54–56]. Polyfunctionality of the T cells and increased numbers of terminally differentiated T cells with a cytotoxic effector potential have been reported in prime-boost vaccine regimens [57, 58]. However, VIA activity is generally not detected after DNA, MVA or canarypox virus prime and protein-boost but may be observed with adenovirus vectors [29, 59].

It is critical that a T cell-based vaccine generates an effective pool of memory CD8[+] T cells that can respond to acute HIV infection or effectively control chronic HIV replication [60]. Our earlier studies reported ADVAX and MVA vaccination-induced both HIV-specific CD4[+] and CD8[+] T cells. In the present study, we demonstrated the induction of CD8[+] T cell-mediated HIV-1 inhibition activity pointing to a potentially protective nature of the CD8[+] T cell response generated by the DNA prime and MVA boost vaccine regimen. Various studies have reported that T$_{SCM}$ cells are critical for the generation of mature CE, EM and TE cells [19, 21, 61, 62]; however, the potential role of these cells in HIV infection and vaccination remains unclear. In the present study, we demonstrated a significant increase in the frequency of both peripheral CD4+ and more prominently CD8[+] T$_{SCM}$ cells following vaccination with a greater increase elicited by the heterologous DNA/MVA prime-boost regimen, thus clearly demonstrating the further scope for testing the prime-boost vaccine regimen.

## Supporting information

**S1 Fig. Representative pseudocolor FACS plot of circulating T stem like memory cells.** CD4[+] T$_{SCM}$ and CD8[+] T$_{SCM}$ cells were gated sequentially on singlets, lymphocyte-sized cells, live CD3[+] T cells, CD4+ T cells, CD8[+] T cells, and then on memory T cells. Central Memory T cells were defined as CD3+CD4+/CD8+CD45RO+CCR7+, Effector Memory cells as CD3 +CD4+/CD8+CD45RO+CCR7-, Terminal Effector cells as CD3+CD4+CD45RO-CCR7- and Naïve T cells as CD3+CD4+/CD8+ CD45RO-CCR7+. T$_{SCM}$ cells were defined as CD3+CD4 +/CD8+CD45RO-CCR7+CD28+CD95+.
(PPTX)

**S1 Table. Commercial reagents used for multicolour flow cytometry.**
(DOCX)

**S2 Table. Frequency of memory CD8+ T cell subsets.**
(DOCX)

**S3 Table. Frequency of memory CD4+ T cell subsets.**
(DOCX)

**S4 Table. Total count of memory CD8+ T cells.**
(DOCX)

**S5 Table. Total count of memory CD4+ T cells.**
(DOCX)

## Acknowledgments

We are thankful to Dr. V. D. Ramanathan who was the Principal Investigator of the P001 trial at the National Institute for Research in Tuberculosis, as well as the volunteers and staff who were part of the trial. We wish to particularly thank the research team for making this study possible through carefully preserving the samples collected during the conduct of the Phase I trial.

The following reagents were obtained through the NIH AIDS Research and Reference Reagent Program, Division of AIDS, NIAID, NIH: HIV-1 ELI from Dr. Jean-Marie Bechet and Dr. Luc Montagnier; HIV-1 IIIB and H9 cell line from Dr. Robert Gallo; TZM-bl from Dr. John C. Kappes, Dr. Xiaoyun Wu and Tranzyme Inc; We thank Professor Johnson Wong and Dr. Galit Alter, Massachusetts General Hospital, Boston, USA, for providing CD3/4 and CD3/8 bi-specific antibodies.

## Author Contributions

**Conceptualization:** Sivasankaran Munusamy Ponnan, Peter Hayes, Soumya Swaminathan, Luke Elizabeth Hanna.

**Data curation:** Sivasankaran Munusamy Ponnan, Natalia Fernandez, Kannan Thiruvengadam, Ashokkumar Srinivasan, Sujitha Kathirvel, Janani Shankar.

**Formal analysis:** Sivasankaran Munusamy Ponnan, Peter Hayes, Natalia Fernandez, Kannan Thiruvengadam, Ashokkumar Srinivasan, Sujitha Kathirvel, Janani Shankar, Rajat Goyal, Nikhil Singla.

**Funding acquisition:** Sivasankaran Munusamy Ponnan, Rajat Goyal, Nikhil Singla, Joyeeta Mukherjee, Shweta Chatrath, Jill Gilmour, Sudha Subramanyam, Srikanth Prasad Tripathy, Soumya Swaminathan, Luke Elizabeth Hanna.

**Investigation:** Sivasankaran Munusamy Ponnan, Natalia Fernandez, Sathyamurthi Pattabiram, Manohar Nesakumar, Ashokkumar Srinivasan, Sujitha Kathirvel, Nikhil Singla, Shweta Chatrath, Jill Gilmour, Sudha Subramanyam, Srikanth Prasad Tripathy, Soumya Swaminathan, Luke Elizabeth Hanna.

**Methodology:** Sivasankaran Munusamy Ponnan, Peter Hayes, Natalia Fernandez, Ashokkumar Srinivasan, Sujitha Kathirvel, Janani Shankar, Nikhil Singla, Joyeeta Mukherjee, Sudha Subramanyam.

**Project administration:** Sivasankaran Munusamy Ponnan, Rajat Goyal, Joyeeta Mukherjee, Shweta Chatrath.

**Resources:** Sivasankaran Munusamy Ponnan, Janani Shankar, Nikhil Singla, Shweta Chatrath.

**Software:** Sivasankaran Munusamy Ponnan, Kannan Thiruvengadam.

**Supervision:** Sivasankaran Munusamy Ponnan, Peter Hayes, Sujitha Kathirvel, Nikhil Singla, Joyeeta Mukherjee, Shweta Chatrath, Luke Elizabeth Hanna.

**Validation:** Sivasankaran Munusamy Ponnan, Peter Hayes.

**Visualization:** Sivasankaran Munusamy Ponnan.

**Writing – original draft:** Sivasankaran Munusamy Ponnan, Sujitha Kathirvel, Luke Elizabeth Hanna.

**Writing – review & editing:** Sivasankaran Munusamy Ponnan, Peter Hayes, Natalia Fernandez, Sujitha Kathirvel, Luke Elizabeth Hanna.

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
