## [Decision Letter · Decision Letter 0]

20 Dec 2019

PONE-D-19-30903

Evaluation of antiviral T cell responses and TSCM cells in volunteers enrolled in a phase I HIV-1 subtype C prophylactic vaccine trial in India

PLOS ONE

Dear Dr. Luke,

Thank you for submitting your manuscript to PLOS ONE. After careful consideration, we feel that it has merit but does not fully meet PLOS ONE’s publication criteria as it currently stands. Therefore, we invite you to submit a revised version of the manuscript that addresses the points raised during the review process.

   Both reviewers had concerns that the data do not in fact support the conclusions as stated. Both reviewers also had concerns with regards to the techniques being utilized and the need for some additional studies to confirm the conclusions reached by the authors. This manuscript therefore requires some additional studies and, in addition, a need to address each of the concerns raised by the 2 reviewers.

We would appreciate receiving your revised manuscript by Feb 03 2020 11:59PM. To enhance the reproducibility of your results, we recommend that if applicable you deposit your laboratory protocols in protocols.io, where a protocol can be assigned its own identifier (DOI) such that it can be cited independently in the future. For instructions see: http://journals.plos.org/plosone/s/submission-guidelines#loc-laboratory-protocols

We look forward to receiving your revised manuscript.

Kind regards,

Aftab A. Ansari, PhD

Academic Editor

PLOS ONE

Journal Requirements:

Reviewers' comments:

Reviewer's Responses to Questions

**Comments to the Author**

1. Is the manuscript technically sound, and do the data support the conclusions?

Reviewer #1: Yes

Reviewer #2: Partly

2. Has the statistical analysis been performed appropriately and rigorously? 

Reviewer #1: Yes

Reviewer #2: No

3. Have the authors made all data underlying the findings in their manuscript fully available?

Reviewer #1: No

Reviewer #2: Yes

4. Is the manuscript presented in an intelligible fashion and written in standard English?

Reviewer #1: Yes

Reviewer #2: Yes

5. Review Comments to the Author

Reviewer #1: In this manuscript Ponnan et al., aims to evaluate an T cell responses and TSCM cells in volunteers enrolled in a phase I HIV-1 subtype C prophylactic vaccine trial in India during 2009-2010. Ponnan et al, hereby report novel findings where they interrogated vaccine induced responses and stem cell memory responses in volunteers who received an HIV-1 subtype C prophylactic vaccine candidate primed with ADVAX DNA and later followed by MVA boost versus either MVA alone or placebo. The results obtained in this study revealed that priming with ADVAX DNA followed with MVA boost versus MVA alone resulted in increased frequencies of peripheral CD8+ TSCM cells and augmented CD8+ T cell-mediated inhibition of replication of different HIV-1 clades. Even-though the manuscript is well written, and these finding may have have significance in the development of vaccines that are urgently needed by high risk populations. However, as stated, the true role of TSCM and actual changes that occur during HIV-1 infection is yet to be fully understood. Therefore, in this manuscript additional investigation is required to confirm the true functionality of CD8+ TSCM cells generated after successful HIV-1 subtype C prophylactic vaccine candidate + ADVAX DNA + MVA boost. Therefore, the manuscript lacks critical experiment such as by sorting CD8+TSCM (a) could they have superior virus inhibition following co-culture with autologous infected CD4+ T cells? (b) in comparison to more differentiated memory phenotypes, are there any differences in cytokine patterns in response to HIV-1 specific overlapping peptides? to make any conclusions on the role of TSCM cells in this prophylactic vaccine trial.

Reviewer #2: In this manuscript, Ponnan and colleagues characterized the phenotype of total CD4 and CD8 T cells, and the ability of in vitro expanded cells to inhibit HIV replication using cells obtained from a HIV vaccine study that was conducted in India many years ago. The authors conclude that cells from vaccinated individuals exhibit some HIV inhibition activity and vaccination induces Tscm cells. Overall, these results are not conclusive and there are some major issues with the analyses as described below.

1) Need to provide details for CD4 T cell infection assay and indicate the ratio of CD4 and CD8 T cells used for virus inhibition assays.

2) Fig 2 – virus inhibition activity is not conclusive. There are no P values and activity was observed even at prevaccination time point (Fig. 2A).

3) Figs 3-6: The data on characterization of different memory CD4 and CD8 T cells are not conclusive and significant for the following reasons. 1) These should be expressed as a percent of total lymphocytes or converted into absolute number to get some idea about the actual change in the number of these cells over time. Currently, they are presented as a percent of total CD4 or CD8 T cells. This is not appropriate. 2) These are total CD4 or CD8 T cells, not vaccine-specific T cells. 3) Where there is an increase, we don’t know if these are MVA or HIV-specific cells. 4) overall, the observed increases are small considering that these are total cells (not antigen-specific).

4) The authors need to correct for multiple comparisons for P values.

6. PLOS authors have the option to publish the peer review history of their article (what does this mean?). If published, this will include your full peer review and any attached files.

Reviewer #1: No

Reviewer #2: No

---

## [Author Response · Author response to Decision Letter 0]

31 Jan 2020

Reviewer #1: In this manuscript Ponnan et al., aims to evaluate an T cell responses and TSCM cells in volunteers enrolled in a phase I HIV-1 subtype C prophylactic vaccine trial in India during 2009-2010. Ponnan et al, hereby report novel findings where they interrogated vaccine induced responses and stem cell memory responses in volunteers who received an HIV-1 subtype C prophylactic vaccine candidate primed with ADVAX DNA and later followed by MVA boost versus either MVA alone or placebo. The results obtained in this study revealed that priming with ADVAX DNA followed with MVA boost versus MVA alone resulted in increased frequencies of peripheral CD8+ TSCM cells and augmented CD8+ T cell-mediated inhibition of replication of different HIV-1 clades. Even-though the manuscript is well written, and these finding may have have significance in the development of vaccines that are urgently needed by high risk populations.

However, as stated, the true role of TSCM and actual changes that occur during HIV-1 infection is yet to be fully understood. Therefore, in this manuscript additional investigation is required to confirm the true functionality of CD8+ TSCM cells generated after successful HIV-1 subtype C prophylactic vaccine candidate + ADVAX DNA + MVA boost. Therefore, the manuscript lacks critical experiment such as by sorting CD8+TSCM (a) could they have superior virus inhibition following co-culture with autologous infected CD4+ T cells? (b) in comparison to more differentiated memory phenotypes, are there any differences in cytokine patterns in response to HIV-1 specific overlapping peptides? to make any conclusions on the role of TSCM cells in this prophylactic vaccine trial.

• We agree with the reviewer that single cell sorting of CD8+ Tscm cells and their analysis for HIV-1 peptide induced cytokine response as well as virus inhibition activity would be extremely interesting. However, we were not in a position to undertake such experiments due to the limited number of PBMC available for this study. As mentioned in the manuscript, this study has been carried out PBMC stored from volunteers who were vaccinated during the period 2009-2010. Using the limited number of cells that were available we studied several aspects of vaccine-induced cellular and humoral immune responses including polyfunctional T cells, T follicular helper cells, Regulatory T cells, memory T cells, Tscm cells, variable loop antibodies, neutralizing antibodies, virus inhibition activity, etc. We have previously published 2 manuscripts in PLOSone entitled “Induction of circulating T Follicular Helper cells and Regulatory T cells correlating with HIV-1 gp120 variable loop antibodies by a subtype C prophylactic vaccine tested in a Phase I trial in India” (https://doi.org/10.1371/journal.pone.0055831) and “Induction and maintenance of bi-functional (IFN-γ + IL-2+ and IL-2+ TNF-α+) T cell responses by DNA prime MVA boosted subtype C prophylactic vaccine tested in a Phase I trial in India” (https://doi.org/10.1371/journal.pone.0213911).

Reviewer #2: In this manuscript, Ponnan and colleagues characterized the phenotype of total CD4 and CD8 T cells, and the ability of in vitro expanded cells to inhibit HIV replication using cells obtained from a HIV vaccine study that was conducted in India many years ago. The authors conclude that cells from vaccinated individuals exhibit some HIV inhibition activity and vaccination induces Tscm cells. Overall, these results are not conclusive and there are some major issues with the analyses as described below.

1) Need to provide details for CD4 T cell infection assay and indicate the ratio of CD4 and CD8 T cells used for virus inhibition assays.

As suggested by the reviewer, we revisited our description of the VIA assay and the suggested details have been incorporated in the manuscript.

2) Fig 2 – virus inhibition activity is not conclusive. There are no P values and activity was observed even at pre vaccination time point (Fig. 2A).

The mean values of log10 reduction was compared at different time points within the placebo, Group A and Group B vaccinees for the 5 different viral clades, using Kruskal-Wallis test followed by subgroup analysis using Dunn’s multiple comparison test to identify differences between the three groups. For all analyses, differences were considered significant if p value was <0.05. The p values have been incorporated in Fig-2.

3) Figs 3-6: The data on characterization of different memory CD4 and CD8 T cells are not conclusive and significant for the following reasons.

• These should be expressed as a percent of total lymphocytes or converted into absolute number to get some idea about the actual change in the number of these cells over time. Currently, they are presented as a percent of total CD4 or CD8 T cells. This is not appropriate.

According to the reviewer's suggestion, we have reanalysed the FACS data and exported absolute count of different subsets of T cells and incorporated as supplementary tables in addition to the percentage frequency of immune cells.

• These are total CD4 or CD8 T cells, not vaccine-specific T cells.

3) Where there is an increase, we don’t know if these are MVA or HIV-specific cells..

4) overall, the observed increases are small considering that these are total cells (not antigen-specific).

We agree with the reviewer that the CD4 and CD8 T cells are not antigen specific. However, we found the significantly increased frequencies of different memory T cells in the vaccine recipients as compared to the placebo recipients. We therefore assume that the increase in immune cell subsets was vaccine induced.

4) The authors need to correct for multiple comparisons for P values.

As suggested by the reviewer we revisited statistical analysis. Since the study includes three groups of volunteers (Placebo, Group-A and Group-B) and different timepoints of vaccination, we used Kruskal-Wallis test to compare vaccine induced responses at different time points and followed it up with subgroup analysis using Dunn’s multiple comparison test to identify differences between the groups. This has been mentioned in the manuscript.

---

## [Editor Report · Decision Letter 1]

3 Feb 2020

PONE-D-19-30903R1

Evaluation of antiviral T cell responses and TSCM cells in volunteers enrolled in a phase I HIV-1 subtype C prophylactic vaccine trial in India

PLOS ONE

Dear Dr Luke,

Thank you for submitting your manuscript to PLOS ONE. After careful consideration, we feel that it has merit but does not fully meet PLOS ONE’s publication criteria as it currently stands. Therefore, we invite you to submit a revised version of the manuscript that addresses the points raised during the review process.

      Please see the minor comments I have posted in the comments to the authors. I would very much like the authors to address thie minor issue. Otherwise, as I stated, the authors have done the best they can to address the issues raised by the 2 reviewers.

We would appreciate receiving your revised manuscript by Mar 19 2020 11:59PM. To enhance the reproducibility of your results, we recommend that if applicable you deposit your laboratory protocols in protocols.io, where a protocol can be assigned its own identifier (DOI) such that it can be cited independently in the future. For instructions see: http://journals.plos.org/plosone/s/submission-guidelines#loc-laboratory-protocols

We look forward to receiving your revised manuscript.

Kind regards,

Aftab A. Ansari, PhD

Academic Editor

PLOS ONE

Additional Editor Comments (if provided):

The authors I believe have done the best they can to address the issues raised by the 2 reviewers. There is only one remaining isue that the authors need to address. In the revised Fig. 2 that appears to be presented out of order, there are only 2 data sets that show asterisk in Fig. 2C ? Are there no other statistically valid differences in the other data presented ? If so, the authors need to modify their manuscript otherwise please edit the figure.

---

## [Author Response · Author response to Decision Letter 1]

6 Feb 2020

Reviewer #1: In this manuscript Ponnan et al., aims to evaluate an T cell responses and TSCM cells in volunteers enrolled in a phase I HIV-1 subtype C prophylactic vaccine trial in India during 2009-2010. Ponnan et al, hereby report novel findings where they interrogated vaccine induced responses and stem cell memory responses in volunteers who received an HIV-1 subtype C prophylactic vaccine candidate primed with ADVAX DNA and later followed by MVA boost versus either MVA alone or placebo. The results obtained in this study revealed that priming with ADVAX DNA followed with MVA boost versus MVA alone resulted in increased frequencies of peripheral CD8+ TSCM cells and augmented CD8+ T cell-mediated inhibition of replication of different HIV-1 clades. Even-though the manuscript is well written, and these finding may have have significance in the development of vaccines that are urgently needed by high risk populations.

However, as stated, the true role of TSCM and actual changes that occur during HIV-1 infection is yet to be fully understood. Therefore, in this manuscript additional investigation is required to confirm the true functionality of CD8+ TSCM cells generated after successful HIV-1 subtype C prophylactic vaccine candidate + ADVAX DNA + MVA boost. Therefore, the manuscript lacks critical experiment such as by sorting CD8+TSCM (a) could they have superior virus inhibition following co-culture with autologous infected CD4+ T cells? (b) in comparison to more differentiated memory phenotypes, are there any differences in cytokine patterns in response to HIV-1 specific overlapping peptides? to make any conclusions on the role of TSCM cells in this prophylactic vaccine trial.

• We agree with the reviewer that single cell sorting of CD8+ Tscm cells and their analysis for HIV-1 peptide induced cytokine response as well as virus inhibition activity would be extremely interesting. However, we were not in a position to undertake such experiments due to the limited number of PBMC available for this study. As mentioned in the manuscript, this study has been carried out PBMC stored from volunteers who were vaccinated during the period 2009-2010. Using the limited number of cells that were available we studied several aspects of vaccine-induced cellular and humoral immune responses including polyfunctional T cells, T follicular helper cells, Regulatory T cells, memory T cells, Tscm cells, variable loop antibodies, neutralizing antibodies, virus inhibition activity, etc. We have previously published 2 manuscripts in PLOSone entitled “Induction of circulating T Follicular Helper cells and Regulatory T cells correlating with HIV-1 gp120 variable loop antibodies by a subtype C prophylactic vaccine tested in a Phase I trial in India” (https://doi.org/10.1371/journal.pone.0055831) and “Induction and maintenance of bi-functional (IFN-γ + IL-2+ and IL-2+ TNF-α+) T cell responses by DNA prime MVA boosted subtype C prophylactic vaccine tested in a Phase I trial in India” (https://doi.org/10.1371/journal.pone.0213911).

Reviewer #2: In this manuscript, Ponnan and colleagues characterized the phenotype of total CD4 and CD8 T cells, and the ability of in vitro expanded cells to inhibit HIV replication using cells obtained from a HIV vaccine study that was conducted in India many years ago. The authors conclude that cells from vaccinated individuals exhibit some HIV inhibition activity and vaccination induces Tscm cells. Overall, these results are not conclusive and there are some major issues with the analyses as described below.

1) Need to provide details for CD4 T cell infection assay and indicate the ratio of CD4 and CD8 T cells used for virus inhibition assays.

As suggested by the reviewer, we revisited our description of the VIA assay and the suggested details have been incorporated in the manuscript.

2) Fig 2 – virus inhibition activity is not conclusive. There are no P values and activity was observed even at pre vaccination time point (Fig. 2A).

As suggested, addition to the Mann Whitney test, mean values of log10 reduction was compared at different time points within the placebo, Group A and Group B vaccines for the 5 different viral clades by post Hoc analysis by using Kruskal-Wallis test followed by subgroup analysis using Dunn’s multiple comparison test to identify differences between the three groups. For all analyses, differences were considered significant if p value was <0.05. We found Group A & B was statistically significant against the placebo only in " U455 viral inhibition at two week post last vaccination".

3) Figs 3-6: The data on characterization of different memory CD4 and CD8 T cells are not conclusive and significant for the following reasons.

• These should be expressed as a percent of total lymphocytes or converted into absolute number to get some idea about the actual change in the number of these cells over time. Currently, they are presented as a percent of total CD4 or CD8 T cells. This is not appropriate.

According to the reviewer's suggestion, we have reanalysed the FACS data and exported absolute count of different subsets of T cells and incorporated as supplementary tables in addition to the percentage frequency of immune cells.

• These are total CD4 or CD8 T cells, not vaccine-specific T cells.

3) Where there is an increase, we don’t know if these are MVA or HIV-specific cells..

4) overall, the observed increases are small considering that these are total cells (not antigen-specific).

We agree with the reviewer that the CD4 and CD8 T cells are not antigen specific. However, we found the significantly increased frequencies of different memory T cells in the vaccine recipients as compared to the placebo recipients. We therefore assume that the increase in immune cell subsets was vaccine induced.

4) The authors need to correct for multiple comparisons for P values.

As suggested by the reviewer we revisited statistical analysis. Since the study includes three groups of volunteers (Placebo, Group-A and Group-B) and different timepoints of vaccination, we used Kruskal-Wallis test to compare vaccine induced responses at different time points and followed it up with subgroup analysis using Dunn’s multiple comparison test to identify differences between the groups. This has been mentioned in the manuscript. 

Additional Editor Comments (if provided):

The authors I believe have done the best they can to address the issues raised by the 2 reviewers. There is only one remaining isue that the authors need to address. In the revised Fig. 2 that appears to be presented out of order, there are only 2 data sets that show asterisk in Fig. 2C ? Are there no other statistically valid differences in the other data presented ? If so, the authors need to modify their manuscript otherwise please edit the figure.

Suggested modification has been carried out.

---

## [Editor Report · Decision Letter 2]

7 Feb 2020

Evaluation of antiviral T cell responses and TSCM cells in volunteers enrolled in a phase I HIV-1 subtype C prophylactic vaccine trial in India

PONE-D-19-30903R2

Dear Dr. Luke,

We are pleased to inform you that your manuscript has been judged scientifically suitable for publication and will be formally accepted for publication once it complies with all outstanding technical requirements.

With kind regards,

Aftab A. Ansari, PhD

Academic Editor

PLOS ONE
---

## [Editor Report · Acceptance letter]

11 Feb 2020

PONE-D-19-30903R2 

Evaluation of antiviral T cell responses and TSCM cells in volunteers enrolled in a phase I HIV-1 subtype C prophylactic vaccine trial in India 

Dear Dr. Hanna:

I am pleased to inform you that your manuscript has been deemed suitable for publication in PLOS ONE. Congratulations! Your manuscript is now with our production department. 

With kind regards,

on behalf of

Dr. Aftab A. Ansari 

Academic Editor

PLOS ONE